# Determinants of the number of days people in the general population spent at home during end-of-life: Results from a population-based cohort analysis

Albert Dalmau-Bueno[1], Anna García-Altés[1,2,3], Jordi Amblàs [4,5]*, Joan Carles Contel[4,5], Sebastià Santaeugènia [4,5]

1 Catalan Agency for Health Quality and Evaluation (AQuAS), Barcelona, Spain, 2 CIBER de Epidemiología y Salud Pública (CIBERESP), Barcelona, Spain, 3 Institut d'Investigació Biomèdica (IIB Sant Pau), Barcelona, Spain, 4 Central Catalonia Chronicity Research Group (C3RG), Centre for Health and Social Care Research (CESS), University of Vic-Central University of Catalonia (UVIC-UCC), Vic, Spain, 5 Chronic Care Program, Department of Health, Barcelona, Spain

* jamblas@chv.cat

**Data Availability Statement:** Owing to the local regulatory framework on data protection, raw data cannot be publicly posted. Data used for this

## Abstract

### Background

The number of days spent at home in the last six months of life has been proposed as a comprehensive indicator of high-value patient-centered care; however, information regarding the determinants of this outcome is scarce, particularly among the general population. We investigated the determinants of spending time at home within the six months preceding death.

### Methods

Population-based, retrospective analysis of administrative databases of the Catalan government. The analysis included adult (≥18 years) individuals who died in Catalonia (North-east Spain) in 2017 and met the McNamara criteria for palliative care. The primary outcome was the number of days spent at home within the last 180 days of life. Other variables included the cause of death, demographic characteristics, and socioeconomic status, stratified as very low, low, mid, and high level.

### Results

The analysis included 40,137 individuals (19,510 women; 20,627 men), who spent a median of 140 days (IQR 16–171) at home within the six months preceding death (women 140 [16–171]; men 150 [100–171]). Female gender was an independent factor of staying fewer days at home (OR 0.80 [95% CI 0.77–0.82]; $p<0.001$). Higher socioeconomic levels were significantly associated with an increasing number of days at home in both genders: among women, ORs of the low, middle, and high levels were 1.09 (0.97–1.22), 1.54 (1.36–1.75), and 2.52 (1.69–3.75) ($p<0.001$), respectively; the corresponding ORs among men were 1.27 (1.12–1.43), 1.56 (1.38–1.77), 2.82 (2.04–3.88) ($p<0.001$). The presence of dementia

analysis are available from the Government of Catalonia Institutional Data Access (PADRIS Program) for researchers who meet the criteria for access to confidential data. Inquiries of access to aggregated data should be addressed to AQuAS director, Ms. Montserrat Moharra (mmoharra@gencat.cat), under the following conditions: http://aquas.gencat.cat/ca/ambits/analitica-dades/padris/.

**Funding:** The authors received no specific funding for this work.

**Competing interests:** The authors have declared that no competing interests exist.

was a strong predictor of spending less time at home in women (0.41 (0.38–0.43); $p<0.001$) and men (0.45 (0.41–0.48); $p<0.001$).

## Conclusions

Our results suggest that end-of-life care is associated with gender and socioeconomic inequalities; women and individuals with lower socioeconomic status spend less time at home within the last 180 days of life.

## Introduction

The increased life expectancy and demographic shift experienced in high-income countries is expected to raise the number of patients in their end-of-life stage [1]. This scenario has encouraged moving towards a patient-centered care approach in which the alignment between patients' preferences and healthcare policies gains importance over classical indicators of health—such as life expectancy—for measuring the quality of healthcare delivery [2]. Nevertheless, evidence suggests that patients' preferences regarding end-of-life care are not always in line with healthcare policies [3–5]. Besides the upset that these inconsistencies may cause on patients and their relatives, some of these practices have high healthcare costs; hence, despite affecting a low proportion of the population, they might compromise the entire system in the advent of the so-called geriatric tsunami [6, 7].

The place of care in the event of a terminal illness is, along with the therapeutic approach (e.g., life-extending vs. palliative), a cornerstone of patients' preferences and healthcare policy regarding end-of-life care. Surveys consistently indicate that most patients with advanced and progressive illnesses prefer to die and spend their last days at home [3, 4]. Although some patients may change these preferences at some point during their end-of-life trajectory, prospective studies suggest that care preferences are rather stable as the illness progresses [8, 9].

The determinants of the place of care during the end-of-life process have been extensively investigated and seem to result from a complex interaction between illness-, individual-, and environmental-related factors [10, 11]. However, most of these studies have been focused on either the place of death or a traditional concept of end-of-life care, typically confined within the last few days of life [12]. Alternatively, it has been recently proposed that the number of days spent at home in the last six months of life might be a more comprehensive indicator of high-value patient-centered care [13]. Since this concept was proposed, few data have become available regarding the determinants of spending more days at home within the last six months [14–17]. The results of these studies—most of them focused on cancer patients—showed heterogeneous results, which are not always aligned with those reported in studies investigating factors influencing end-of-life. In this analysis, we have investigated the factors that influence staying at home within the six months preceding death in the general population.

## Materials and methods

### Study design

This was a population-based, retrospective analysis of end-of-life trajectories of all adult (i.e., 18 years or older) individuals who died in Catalonia (North-east Spain) in 2017 due to advanced and progressive illness. Individuals were selected from the mortality registry of the Catalonian Ministry of Health; the analysis included all individuals of the registry whose cause

of death met the diagnostic criteria proposed by McNamara et al. to define palliative care populations [18].

The Catalan Health Service provides universal and free healthcare (including primary, hospital, and intermediate care) to the entire population. The sole exception is drug prescription, which is based on a co-payment system calculated according to the individual's income [19]. Long-term care in our area is provided by public and private (nearly half) facilities, including nursing homes and mental health facilities for people with severe mental disorders. Each citizen is assigned a unique personal healthcare identification code, which was used to gather information regarding the socioeconomic and clinical characteristics of these individuals, as well as the use of social and healthcare resources during the 180 days preceding death.

This was a retrospective analysis of data anonymously collected in the setting of routine management by the national healthcare provider; no data from hospital electronic health records were included. Since the analysis did not involve any intervention, the Agency for Health Quality and Assessment of the Catalan Minstry of Health waived ethical approval for analysis conduct. Analyzed data were recorded in various central databases of the Catalan Ministry of Health between January and December 2017, and extracted on March 2020. All data were handled according to the General Data Protection Regulation 2016/679 on data protection and privacy for all individuals within the European Union and the local regulatory framework regarding data protection.

## Data sources

Clinical and healthcare services use information was retrieved from the healthcare registry of the Catalan Ministry of Health (CMBD, *Conjunt Mínim Bàsic de Dades*). The Catalan Health Service provides universal healthcare to the entire population (i.e., 7.5 million people). The CMBD systematically collects information regarding resource utilization, including acute care hospitals, intermediate and long-term care, emergency services, primary care, and mental health centers. The CMBD has an automated data validation system that checks data consistency and identifies potential errors. Furthermore, as this information is used for provider payment purposes, external audits are regularly conducted to ensure the quality and reliability of the data [20, 21]. All datasets are linked through a unique insurance number. Further details on the datasets retrieved from the CMBD are listed in the S1 File.

Data regarding pharmaceutical prescription were retrieved from the pharmacy activity registry, which also identifies individuals institutionalized in a health and social care facilities for intermediate and long-term care (irrespective of whether they are private or public), as well as non-institutionalized individuals. Demographic and socioeconomic information were retrieved from the central population registry [22]. Finally, deaths—and the main cause—and availability of advanced directives (living wills) were identified from the mortality registry of Catalonia and the advanced directives registry of the Catalan Ministry of Health.

## Variables and outcomes

The primary study outcome was the percentage of days spent at home within the last 180 days of life. Other environments included nursing homes, acute care hospitals, emergency departments, emergency medical services, out-of-hours urgent primary care services, and intermediate care facilities, either for rehabilitation or palliative care purposes. The stay of the individual in all settings except home was directly retrieved from the various CMBDs or the pharmacy registry level; time spent at home was defined as the complementary outcome of not being in any other environment.

Characteristics of study participants included gender, age at death, the cause of death, citizenship, socioeconomic status (very low, low, mild, and high, based on annual income categories defined for pharmaceutical copayment), living alone, having completed their advanced directives, and being a recipient of a specific local healthcare program: domiciliary care, palliative care, complex chronic patient (PCC, *Pacient Crònic Complex*), and advanced chronic disease (MACA, *Malaltia Crònica Avançada)* (S1 File provides further details on the variables collected).

## Analyses

The characteristics of the study population and causes of death were described according to gender and reported using absolute and relative frequencies. The number of days spent in each place was first described using measures of central tendency (mean and median) and the corresponding measures of dispersion (standard deviation [SD] and interquartile range [IQR], defined by the 25th and 75th percentiles).

To determine the factors associated with spending more days at home during the last 180 days, we carried out a robust generalized multivariate linear model with a logit link and the binomial family, with the percentage of days spent at home as a dependent variable. The analysis of the determinants of spending time at home environment was performed for sociodemographic, clinical variables, and causes of mortality separately (unadjusted), and by adjusting for all these factors. Results were presented as the estimated percentage of days spent at home and the odds ratio (OR) associated with each factor, along with the corresponding 95% confidence interval (95% CI) and its corresponding *p*-value. All analyses were performed for each gender separately. Complementarily, the relevance of gender in spending more days at home during the 180 days was explored by including this variable in a fully robust adjusted generalized model. The significance threshold was set at a two-sided alpha value of 0.05 for all analyses, which were performed with the Stata Program, version 14.2 (UCLA: Statistical Consulting Group).

## Results

### Characteristics of study patients

The analysis included 40,137 individuals (19,510 women and 20,627 men) aged 18 years or more who died in Catalonia in 2017 and met the McNamara criteria for palliative care (S1 Fig in S1 File, S1 File). Table 1 summarizes the demographic, socioeconomic, and clinical characteristics of women and men included in the analysis. Most individuals (94% of women and 95% of men) had an annual income below 18,000€ or between 18,000€ and 100,000€. Cancer (35%) and dementia (27%) were the leading causes of death among women, whereas men more frequently died from cancer (51%), followed by and COPD (15%). Living alone was more frequent among women (12%) than men (5%).

### Days spent at home

Overall, the study population spent a median of 140.0 days at home (IQR 16.0–171.0) (mean 106.8; SD 71.5) (S2 Fig in S1 File). Fig 1 summarizes the proportion of individuals of the study population staying at home and the investigated facilities along the last 180 days of life. The number of individuals staying at home progressively decreased throughout the entire period and dropped rapidly within the last 60 days. Correspondingly, the number of individuals staying in the hospital increased progressively as approaching the end of life. Fig 2 shows the transitions in the place of stay that occurred in the last ten days of life (i.e., day 171 of the

**Table 1. Demographic, socioeconomic, and clinical characteristics of the study population.**

| | Women | Men |
|---|---|---|
| | (n = 19,510) | (n = 20,627) |
| | No. (%) | No. (%) |
| **Age (years)** | | |
| <55 | 843 (4.32) | 1173 (5.69) |
| 55–64 | 1179 (6.04) | 2269 (11) |
| 65–69 | 865 (4.43) | 1781 (8.63) |
| 70–74 | 1103 (5.65) | 2182 (10.58) |
| 75–79 | 1445 (7.41) | 2508 (12.16) |
| 80–84 | 3054 (15.65) | 3836 (18.6) |
| 85–89 | 4508 (23.11) | 3788 (18.36) |
| 90–95 | 4301 (22.05) | 2306 (11.18) |
| ≥95 | 2212 (11.34) | 784 (3.8) |
| **Citizenship** | | |
| Spanish | 19033 (97.56) | 20032 (97.12) |
| Foreign | 477 (2.44) | 595 (2.88) |
| **Socioeconomic level** | | |
| Recipients of social services[1] | 873 (4.47) | 708 (3.43) |
| Annual income <18,000€ | 15448 (79.18) | 14352 (69.58) |
| Annual income 18,000€ -100,000€ | 2960 (15.17) | 5285 (25.62) |
| Annual income >100,000€ | 89 (0.46) | 123 (0.6) |
| Not available | 140 (0.72) | 159 (0.77) |
| **Cause of death** | | |
| Tumors | 6833 (35.02) | 10417 (50.5) |
| Heart failure | 3588 (18.39) | 2589 (12.55) |
| Kidney disease | 675 (3.46) | 542 (2.63) |
| Liver failure | 222 (1.14) | 492 (2.39) |
| COPD | 2123 (10.88) | 3179 (15.41) |
| Nervous system | 737 (3.78) | 846 (4.1) |
| Dementia | 5316 (27.25) | 2518 (12.21) |
| AIDS | 16 (0.08) | 44 (0.21) |
| **Advanced directives available** | 77 (0.39) | 107 (0.52) |
| **Living alone** | 2324 (11.91) | 1088 (5.27) |
| **Recipient of specific care programs[2]** | | |
| Complex chronic patient | 4492 (23.02) | 4007 (19.43) |
| Advanced chronic disease | 5147 (26.38) | 4674 (22.66) |
| Domiciliary care | 6716 (34.42) | 4868 (23.6) |
| Palliative care | 2196 (11.26) | 2145 (10.4) |

[1]Includes individuals perceiving a minimum integration income, unemployment allowance, unemployment benefit or not qualifying for either of the previous.

[2]Categories are not mutually exclusive.

investigated period). At that time point, most individuals were at home (n = 15,441; 38.47%), followed by acute care hospital (n = 7,760; 19.63%), and nursing homes (n = 6,384; 15.91%). From that time point on, most transitions occurred from home to acute care hospital, except the last day before death, when nearly half of the patients who left home moved to emergency services/department. Conversely, individuals institutionalized in nursing homes showed a

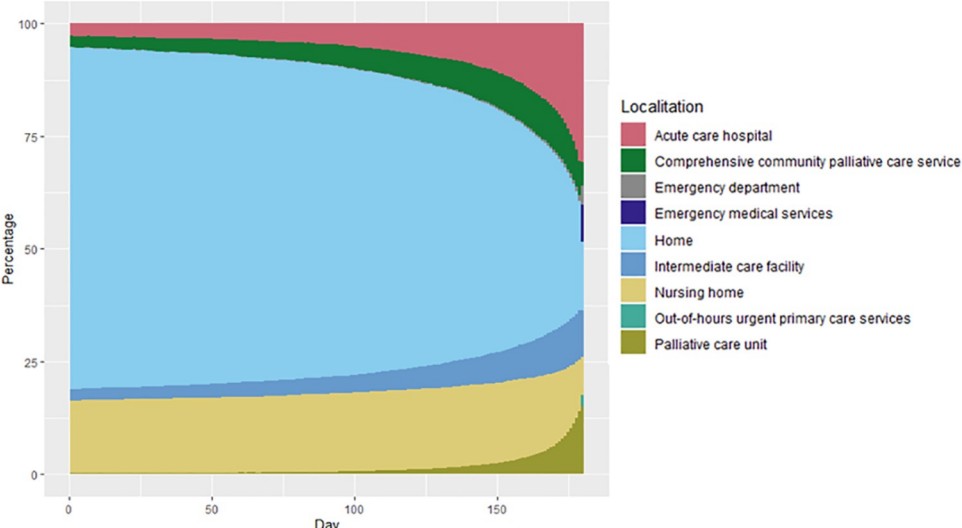

**Fig 1. Percentage of patients staying at home and at each facility within the last six months of life.** S1 Fig in S1 File shows the distribution of cases across the primary variable (i.e., days spent at home within the last 180 days of life).

more heterogeneous transfer destination, including hospital, comprehensive community palliative care services, and home, with no dominance of a particular destination. Although most individuals admitted to acute care hospitals remained in this place until death, we observed some transitions to home within the few last days of life.

Overall, women and men spent a mean (SD) of 106.8 (71.5) and 125.1 (60.0) days at home in the last 180 days of their life, respectively (Table 2). The number of days spent at home progressively declined with age and raised with the socioeconomic status of both women and men. The overall trend of men spending more time than women at home became evident across all sociodemographic characteristics. For instance, women aged <55 years spent more time at home than those aged 90–95 years (mean of 133.2 and 97.4 days, respectively); the

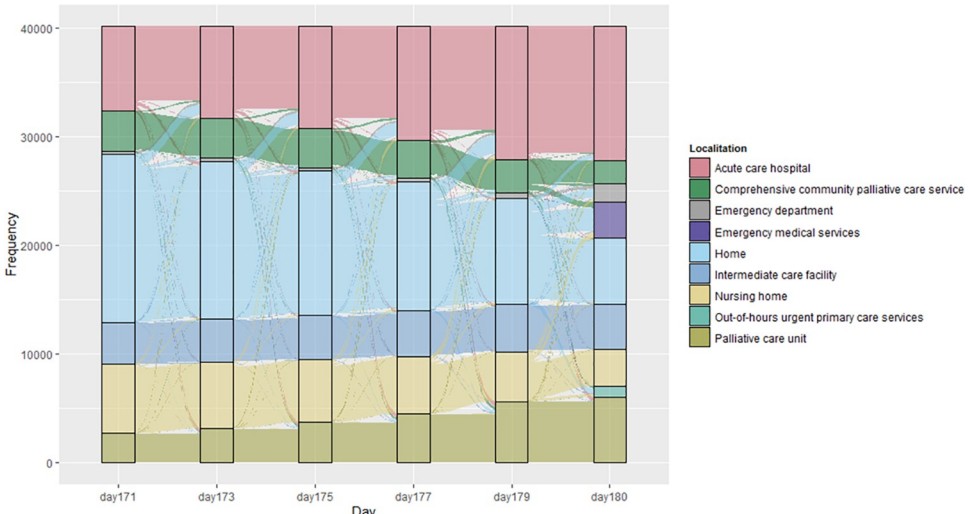

**Fig 2. Transition of place of stay during the last 10 days before death.** Vertical boxes show the place of stay at the given day, counting from 180 days before death.

**Table 2. Descriptive analysis of the number of days spent at home within the last 180 days of life among women and men, according to clinical and socioeconomic characteristics.**

| | Women | | Men | |
|---|---|---|---|---|
| | **Mean (SD)** | **Median (IQR)** | **Mean (SD)** | **Median (IQR)** |
| **Total** | 106.77 (71.52) | 140.00 (16.00–171.00) | 125.10 (60.00) | 150.00 (100.00–171.00) |
| **Age (years)** | | | | |
| <55 | 133.22 (49.90) | 150 (113.00–170) | 143.12 (44.06) | 157.00 (131.00–176.00) |
| 55–64 | 135.75 (47.05) | 152.00 (118.00–169.00) | 135.86 (47.94) | 151.00 (123.00–170) |
| 65–69 | 126.13 (55.72) | 147.00 (106.00–168.00) | 130.70 (51.73) | 149.00 (115.00–168.00) |
| 70–74 | 121.20 (59.98) | 146.00 (91.00–168.00) | 130.62 (53.19) | 150 (113.00–169.00) |
| 75–79 | 115.82 (63.45) | 141.00 (74.00–168.00) | 125.39 (56.89) | 147.00 (103.00–168.00) |
| 80–84 | 107.84 (69.56) | 138.00 (30–168.00) | 122.78 (60.75) | 149.00 (92.00–170) |
| 85–89 | 100.87 (73.98) | 133.00 (1.00–170) | 118.83 (66.24) | 150 (72.00–172.00) |
| 90–95 | 97.37 (77.08) | 132.00 (0–173.00) | 114.38 (70.27) | 149.50 (37.00–174.00) |
| ≥95 | 89.40 (80.06) | 105.50 (0–174.00) | 111.24 (73.34) | 150 (15.50–176.00) |
| **Citizenship** | | | | |
| Spanish | 106.20 (71.79) | 139.00 (12.00–170) | 124.75 (60.27) | 150 (99.00–171.00) |
| Foreign | 124.29 (56.33) | 145.00 (102.00–168.00) | 132.59 (50.52) | 148.00 (118.00–169.00) |
| Not available | 145.17 (48.11) | 167.50 (126.00–179.00) | 150.03 (40.42) | 164.00 (139.50–179.00) |
| **Socioeconomic status** | | | | |
| Recipients of social services* | 106.98 (68.19) | 135.00 (31.00–167.00) | 121.39 (61.53) | 148.00 (93.00–170) |
| <18.000€ | 103.13 (72.57) | 135.00 (5.00–170) | 122.42 (61.29) | 149.00 (93.00–170) |
| 18.000€ -100.000€ | 123.10 (64.93) | 153.00 (88.00–175.00) | 131.50 (56.22) | 154.00 (114.00–173.00) |
| >100.000€ | 142.49 (54.64) | 168.00 (132.00–179.00) | 152.09 (40.25) | 171.00 (142.00–179.00) |
| Not available | 139.29 (52.91) | 164.00 (115.50–179.00) | 149.70 (39.90) | 163.00 (139.00–179.00) |
| **Cause of death** | | | | |
| Tumors | 124.43 (56.30) | 146.00 (101.00–166.00) | 130.32 (51.19) | 149.00 (114.00–167.00) |
| Heart failure | 118.89 (69.82) | 154.50 (54.00–177.00) | 140.48 (56.49) | 167.00 (131.00–179.00) |
| Kidney disease | 115.88 (70.50) | 154.00 (33.00–174.00) | 128.90 (59.16) | 156.00 (106.00–174.00) |
| Liver failure | 121.38 (57.09) | 141.00 (97.00–166.00) | 139.23 (45.79) | 155.00 (124.00–171.00) |
| COPD | 104.92 (74.31) | 141.00 (2.00–173.00) | 122.21 (63.59) | 150 (90–173.00) |
| Nervous system | 98.84 (77.89) | 135.00 (0–175.00) | 116.70 (70.96) | 156.00 (40–176.00) |
| Dementia | 75.91 (78.14) | 32.00 (0–168.00) | 90.63 (75.79) | 109.50 (0–168.00) |
| AIDS | 124.75 (59.03) | 140 (99.00–176.50) | 124.02 (53.95) | 138.00 (105.50–164.50) |
| **Living wills available** | | | | |
| Yes | 128.34 (54.43) | 149.00 (103.00–170) | 121.75 (57.18) | 145.00 (93.00–165.00) |
| No | 106.68 (71.57) | 140 (15.00–171.00) | 125.12 (60.01) | 150 (100–171.00) |
| **Living alone** | | | | |
| Yes | 99.66 (74.26) | 131.00 (0–169.00) | 115.84 (65.97) | 146.00 (64.00–170) |
| No | 108.90 (70.54) | 141.00 (28.00–171.00) | 127.34 (58.25) | 151.00 (106.00–171.00) |

*Includes individuals perceiving a minimum integration income, unemployment allowance, unemployment benefit or not qualifying for either of the previous nor other categories.

The number of days spent in other resources is summarized in S1 Table in S1 File (women) and S2 Table in S1 File (men) (S1 File)

corresponding times for men at the same age groups were 143.1 and 114.4, respectively. Similarly, women in the lowest socioeconomic status spent fewer days at home than those with higher annual income than 100,000 € (mean 107.0 vs. 142.5), while the corresponding values for men were 121.4 and 152.1, respectively. Finally, Spanish women tended to stay less time at

home than foreign women (mean 106.2 vs. 124.3 days); men showed a similar trend, although with systematically higher time than women (132.6 vs. 124.8).

Regarding the cause of death, individuals who died from dementia stood fewer days at home in both genders. The gender imbalance was also pervasive in this factor, with women and men who died from dementia spending a mean of 75.9 and 90.6 days at home, respectively, and those who died from cancer 124.4 and 130.3.

## Factors influencing time spent at home

The analysis of the percentage of days spent at home for the entire population showed that women had lower odds to stay at home (adjusted OR 0.80 [95% CI 0.77–0.82]; $p<0.001$) (S3 Table in S1 File). Fig 3 summarizes the results of the adjusted model for men and women. In both genders, higher age and lower socioeconomic status were associated with spending less time at home. Compared with women younger than 55 years, those in the age group of 90–95 spent 36% less time at home (OR 0.65; 95% CI 0.55–0.72); the corresponding decrease among men was 44% (OR 0.56; 0.50–0.63). Regarding the socioeconomic status, women perceiving more than 100,000 € a year had more than twice higher odds to spend time at home compared with those in the lower stratum of socioeconomic status (OR 2.52 [95% CI 1.69–3.75]); the corresponding OR for men was 2.82 (2.04–3.88). The unadjusted analysis showed the same trends as the adjusted analysis for age, gender, and socioeconomic status (S4 Table in S1 File).

In both genders, compared with tumors, dying from dementia was a strong predictor of spending less time at home (OR 0.41 [95% CI 0.38–0.43] in women and 0.45 [0.41–0.48] in men). COPD and nervous system diseases were also associated with less time at home, albeit

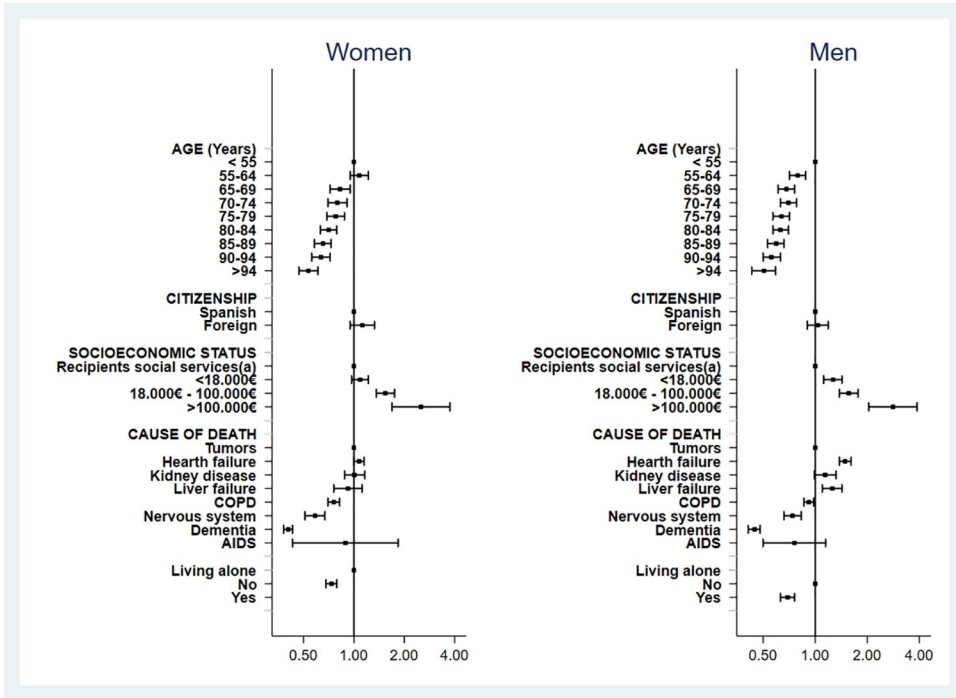

**Fig 3. Adjusted model of percentage of days spent at home within the last 180 days of life, according to gender.** Squared dots and lines show the odds ratio and 95% confidence interval of spending a higher percentage of days at home compared with the reference category. The model has been adjusted for all investigated variables. (a) Includes individuals perceiving a minimum integration income, unemployment allowance, unemployment benefit or not qualifying for either of the previous nor other categories.

to a lesser extent. In men, dying from heart and liver failure were significantly associated with spending more time at home. In women, none of these diagnoses significantly contributed to the time spent at home in the adjusted analysis (although unadjusted analysis showed a significant influence). Living alone was associated with spending less time at home in both men and women.

## Discussion

In this retrospective analysis of a population-based cohort, we found that older people who died from an advanced and progressive illness spent nearly a quarter of the last six months of life in care facilities. The frequency of transition from home to these facilities became particularly frequent within the 60 days preceding death. Male gender and high socioeconomic status were significantly associated with spending more time at home. The cause of death also influenced the place of stay, with dementia—more prevalent among women—being a strong predictor of spending less time at home. Individuals who lived alone were less likely to spend time at home during the last six months of life.

While a body of evidence has been published regarding the determinants of the place of death, to our knowledge, only four studies have specifically investigated the factors that influence time spent at home within the last six months of life: three in population-based cohorts of cancer patients and one in a community-based cohort of 754 patients [14–17]. The population in our analysis was selected using the McNamara criteria for defining palliative care populations in the research setting [18]. Of note, the number of patients eligible for palliative care is typically higher than those receiving it—as was the case of our study sample, with only 10% of individuals receiving palliative care [23, 24]. Hence, rather than palliative care recipients, the population included in our analysis corresponded to individuals who died from advanced and progressive illness eligible for planned end-of-life care. Overall, the median time spent at home in our study population (140 days) was on the lower side of the range found in previous studies (156 to 164 days, both extremes corresponding to cancer populations) [14–16]. Patients who died from cancer in our cohort tended to spend more days at home, though the average time remained below that of previous studies on cancer patients.

Gender came up as an important determinant of staying at home during the last six months, with men showing persistently higher percentages of days at home across all variables analyzed. This trend, consistent with previous reports of population-based cohorts of cancer patients [15, 16], illustrates how gender inequalities in healthcare provision observed elsewhere [25] persist during the end-of-life trajectory. In line with the sustainable development goals of the WHO [26], the Catalan government urged to explicitly incorporate a gender perspective in all phases of health planning [27]. Hence, we investigated the factors influencing the amount of time spent at home in women and men separately. The mechanisms underlying gender inequalities are complex and may overlap with other factors such as longevity (women in our population tended to be older), socioeconomic status (women tended to receive lower annual incomes), and social environment (living alone was twice more frequent among women than men), among others. Still, our findings reinforce the need for incorporating a gender perspective in the design of health policies regarding end-of-life trajectory and care programs for patients with advanced diseases.

Another important contributor to spending more time at home within the last six months of life was the socioeconomic status, with higher income associated with more days at home in both genders. A similar trend was reported by Anderson et al., although they found a more modest contribution of this factor [14]. The type of population (i.e., cancer patients vs. community-based population) may have contributed to the different weight of this factor.

However, it is worth mentioning that Anderson et al. stratified individuals by income quintiles, whereas we used the strata established for drug reimbursement purposes, which result in more extreme differences between income categories and may affect the sensitivity of the analysis. Regardless of the weight of the influence of the socioeconomic status on time spent at home, our analysis depicts a clear gradient across socioeconomic categories. Like in the case of gender, this finding indicates that the socioeconomic inequities found in other healthcare indicators (e.g., emergency care visits and hospitalization rate) [28] among older persons also persist in end-of-life pathways. The impact of socioeconomic status on multiple aspects of care reinforces the need for designing integrated health and social care strategies.

In addition to the patient- or disease-related characteristics, the likelihood of staying at home might be reasonably influenced by context-specific factors such as the organization of the healthcare system. In our environment, patients with chronic and advanced diseases may be eligible for specific care programs aimed at providing multidisciplinary, patient-centered care. However, interactions between these programs (i.e., one patient may be a recipient of two or more care programs simultaneously) challenge the analysis and interpretation of the specific contribution of each of them to the percentage of days spent at home. Future studies designed *ad hoc* to investigate the influence of each of these programs and their combination on this outcome are warranted.

Our results are strengthened by the population-based approach, which allowed us to gather basic socioeconomic and clinical information (both, primary care and specialized) from all individuals covered by the national system of health—virtually the entire population, considering the universal coverage of the healthcare system in our country. On the other hand, retrospective studies based on administrative databases are constrained by the number and type of variables recorded. In this regard, we could not adjust for disease severity (i.e., the CMBD records the diagnostic code only), and cultural factors (e.g., religious, relationship patterns, etc.) with known influence on patients' preferences and behaviors regarding end-of-life trajectories [29]. Likewise, the availability of advanced directives could be identified in less than 1% of our sample, and family preferences regarding end-of-life care, which strongly influences the place of stay [11], could not be assessed. Another constrain regarding the data sources was the predefined cutoffs of socioeconomic status, which are tailored to classy the population into levels of pharmaceutical co-payment. Nevertheless, this approach has allowed identifying socioeconomic differences in other healthcare outcomes, with consistent results across studies [28, 30, 31]. Finally, studies investigating healthcare outcomes in end-of-life trajectory are challenged by the boundaries of the population to be investigated in this setting [18, 23, 24]. We considered the McNamara criteria, based on the ICD-10 causes of death, the most suitable system for selecting patients with advanced and progressive illnesses, who could be, therefore, candidates to planned end-of-life care.

## Conclusions

Our results show how despite some efforts recently implemented to move towards patient-centered care, individuals with chronic, advanced, and progressive illnesses spend a remarkable amount of time in health and social care facilities within the last six months of life. More worrisome, the significant influence of gender and socioeconomic status on this outcome suggests that gender and social inequalities often observed in healthcare may also affect the end-of-life trajectory. These characteristics (and with living alone) should drive prioritization of end-of-life care policies. Taken together, these findings encourage including the outcome "days spent at home during the last six months of life" in healthcare programs as a quality indicator of patient-centered care. The inclusion of this outcome may contribute to designing

integrated health and social care pathways flexible enough to provide adequate care during the end-of-life stage. This approach might also offer a more comprehensive perspective of care in this stage of life that goes beyond efficiency indicators such as cost, hospitalization rates, or admissions to the emergency department. Future works to investigate how these factors behave in each cause of death are warranted.

## Supporting information

**S1 File.**
(PDF)

## Acknowledgments

The authors would like to thank Cristina Colls, Raul González, and Ingrid Bullich for their helpful insights on the study conception. Gerard Carot-Sans provided medical writing assistance during the manuscript preparation.

## Author Contributions

**Conceptualization:** Albert Dalmau-Bueno, Anna García-Altés, Jordi Amblàs, Joan Carles Contel, Sebastià Santaeugènia.

**Formal analysis:** Albert Dalmau-Bueno.

**Investigation:** Anna García-Altés, Jordi Amblàs, Joan Carles Contel, Sebastià Santaeugènia.

**Methodology:** Albert Dalmau-Bueno.

**Supervision:** Anna García-Altés.

**Writing – original draft:** Albert Dalmau-Bueno, Anna García-Altés, Jordi Amblàs, Joan Carles Contel, Sebastià Santaeugènia.

**Writing – review & editing:** Albert Dalmau-Bueno, Anna García-Altés, Jordi Amblàs, Joan Carles Contel, Sebastià Santaeugènia.

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
