## [Decision Letter · Decision Letter 0]

15 Mar 2021

PONE-D-20-40096

Determinants of the Number of Days Spent at Home During End-of-Life Trajectory among General Population: Results from a Population-Based Cohort Analysis

PLOS ONE

Dear Dr. Amblàs,

Thank you for submitting your manuscript to PLOS ONE. After careful consideration, we feel that it has merit but does not fully meet PLOS ONE’s publication criteria as it currently stands. Therefore, we invite you to submit a revised version of the manuscript that addresses the points raised during the review process.

We look forward to receiving your revised manuscript.

Kind regards,

Rosa Maria Urbanos Garrido, PhD

Academic Editor

PLOS ONE

2. In the ethics statement in the manuscript and in the online submission form, please provide additional information about the patient records used in your retrospective study, including:

a) whether all data were fully anonymized before you accessed them;

b) the date range (month and year) during which patients' medical records were accessed;

c) the date range (month and year) during which patients whose medical records were selected for this study sought treatment; and

d) the source of the medical records analyzed in this work (e.g. hospital, institution or medical center name).

If patients provided informed written consent to have data from their medical records used in research, please include this information.

3. Please ensure you have thoroughly discussed any potential limitations of this study within the Discussion section, including the potential impact of confounding factors.

5. Please include captions for your Supporting Information files at the end of your manuscript, and update any in-text citations to match accordingly. Please see our Supporting Information guidelines for more information: http://journals.plos.org/plosone/s/supporting-information

Reviewers' comments:

Reviewer's Responses to Questions

**Comments to the Author**

1. Is the manuscript technically sound, and do the data support the conclusions?

Reviewer #1: Partly

Reviewer #2: Yes

Reviewer #3: Yes

2. Has the statistical analysis been performed appropriately and rigorously? 

Reviewer #1: N/A

Reviewer #2: Yes

Reviewer #3: No

3. Have the authors made all data underlying the findings in their manuscript fully available?

Reviewer #1: No

Reviewer #2: Yes

Reviewer #3: No

4. Is the manuscript presented in an intelligible fashion and written in standard English?

Reviewer #1: Yes

Reviewer #2: Yes

Reviewer #3: No

5. Review Comments to the Author

Reviewer #1: The paper investigates a policy-relevant question which can be translated into important policy recommendations in the end-of-life care. Also, the authors use a rich dataset. The addressed subject is interesting and relevant, and the paper is interesting to read, however I have some comments and concerns that I detail below.

I think my main point is the fact that the outcome variable appears to show a bimodal distribution. In view of this condition, the standard generalized linear model with a binomial error distribution often demonstrates insufficient predictive performance when analyzing proportional data.

Histograms for the outcome variable must be provided. In light of the results of descriptive statistics and Figure 1, I suspect that the response variable appear to follow a bimodal distribution, accumulating many observations in both tails.

To address this difficulty, I propose a beta regression model with shape parameters close to 0.5 or an asymmetric logistic regression model that uses a new parameter to account for data complexity.

Second, authors should provide more information to try to convince the reader that the socioeconomic status (based on annual income categories defined for pharmaceutical copayment), has sufficient explanatory power since it accumulates most of its observations in the annual income <18,000€ interval.

Reviewer #2: The manuscript seems to have used sound methodology and the results and conclusions were presented in a clear way. Study limitations were also dully acknowledged. I have no suggestions or corrections to propose.

Reviewer #3: Dear authors.

I think this is an interesting paper with a large potential. It focuses on the number of days that people in the general population (selected based on McNamara) in Catalonia, Spain, spend at home during their end-of-life, defined as 6 months before death. The paper is policy relevant and contributes nicely to a scarce literature on end-of-life trajectories.

The authors find that individuals with chronic, advanced, and progressive illnesses spend a remarkable amount of time in health and social care facilities within the last six months of life. They also find that gender and socioeconomic status influences this outcome, and therefore suggest that gender and social inequalities, often observed in healthcare provision, may also affect the end-of-life trajectory.

The paper is well written; however, it would benefit from having an English edit. The setting (i.e., how healthcare is provided in Catalonia), material (how and why registers are informed with information, and their coverage rate) is not explained well enough. The methods are quite easy to follow, however, I would prefer that the outcome of all analyses is ‘days at home’. Also, there should be a clear distinction between descriptive statistics, and analyses where the authors have predicted how factors influence the number of days spent at home while controlling for other factors.

Below, I give a point by point list of comments to the manuscript:

1. Title: I do not like the wording in the title. Maybe the title could be changed to: “Determinants of the number of days people in the general population spent at home during their end-of-life: results from a population-based cohort analysis”

2. I think the language in the abstract can be approved. For example, in the sentence: “The analysis included adult (i.e., ≥18 years) individuals died in Catalonia in 2017, who met the McNamara criteria for palliative care” I would remove ‘i.e.’, and included ‘who’: “The analysis included adult (≥18 years) individuals who died in Catalonia in 2017, who met the McNamara criteria for palliative care”.

Similarly, several other places, I think the manuscript can be improved by editing the language.

3. In the abstract, it is not intuitive to understand what the authors refer to when they indicate the results from their analyses – since the methods have not been explained. For example: “Higher socioeconomic levels were significantly associated with an increasing number of days at home in both genders: among women, ORs of the second, third, and fourth levels were 1.09 (0.97–1.22), 1.54 (1.36–1.75), and 2.52 (1.69–3.75) (p<0.001), respectively; the corresponding ORs among men were 1.27 (1.12–1.43), 1.56 (1.38–1.77), 2.82 (2.04–3.88) (p<0.001).”

What does the second, third and fourth level refer to? No levels have been introduced to the reader. Either the methods section needs to be expanded – or the results should not refer to numbers and analyses the reader can not understand how were calculated. Also, because methods have not been explained, it is impossible to understand whether the analyses in the abstract have been controlled for confounding factors or whether they are purely descriptive analyses.

As I will point out later in my comments, I think results should always be presented as ‘days at home’ – not odds ratios. This would greatly ease the read for the audience.

4. The conclusion in the abstract is strange, since it states that “the presence of gender”. Since we all have a gender, you should refer to ‘females’ or ‘males’, or that gender (not the presence of gender) influenced the results.

5. Study design – overall – I think the material and methods are not explained well enough. See points below for details.

6. It should be stated clearly whether all adults who died in Catalonia were included?

7. It must be elaborated what the authors have done to select the patients – what does it mean that they have used the framework by McNamara? Who have then been included? Maybe provide a patient flow?

8. The authors should explain the healthcare system in Catalonia (public versus private, funding – tax or out of pocket) for the reader to understand the system where analyses have taken place.

9. The authors should give a more detailed description of the data – how and why are they gathered in the registries. Do they include all healthcare services that has been provided (also privately provided services – if available)? Some of the information in the appendix can be elaborated and moved to the main manuscript?

10. Why do you use a generalized linear model with a logit link and a binomial family – how did you do the model selection? I would like to know how robust these analyses are – meaning – how much are the results influenced by specifying the model in a different way?

11. In the methods, what does ‘each category’ refer to?

12. I think it would be a good idea to use ‘days spent at home’ as the outcome measure, also in the generalized linear models, and to select an appropriate model from among possible models, for example, the poisson, the negative binomial model, two-part models, zero-inflated models etc. See for example: Deb P, Norton EC, Manning WG. Health Econometrics Using Stata. 1 ed. Texas: Stata Press. 2017.

13. I really like Figure 1! Figure 2 is not possible for me to read (the picture is not clear). However, the idea of seeing where people move from and to is intriguing, so I encourage a figure showing transition.

14. In the results where authors write “Overall, women and men spent a mean (SD) of 106.8(71.5) and …” it is not clear to me whether this is non-adjusted or adjusted analyses. It needs to be stated clearly!

15. The analyses of factors influencing time spent at home should be analyzed with the output ‘days spent at home’. For me, it is not intuitive to understand how the OR and percentage of days spent at home should be interpreted. However, if you give results in the no. of days spent at home (with reference either to a reference category, or, as the total no. of days) it would be much easier to read.

16. The authors should elaborate on what changes when they run crude and fully controlled analyses.

17. Is there a difference pattern of end-of-life trajectories between those dying from different causes – can the analyses be stratified by cause of death (as it has been by gender)?

6. PLOS authors have the option to publish the peer review history of their article (what does this mean?). If published, this will include your full peer review and any attached files.

Reviewer #1: No

Reviewer #2: No

Reviewer #3: No

---

## [Author Response · Author response to Decision Letter 0]

14 May 2021

A document containing the same responses to reviewers' comments has been uploaded with ms documents (end of pdf proof, after revised manuscript).

Responses to reviewers’ comments

Reviewer 1

The paper investigates a policy-relevant question which can be translated into important policy recommendations in the end-of-life care. Also, the authors use a rich dataset. The addressed subject is interesting and relevant, and the paper is interesting to read, however I have some comments and concerns that I detail below.

RESPONSE: we thank the reviewer for his/her comments and interest in our work.

I think my main point is the fact that the outcome variable appears to show a bimodal distribution. In view of this condition, the standard generalized linear model with a binomial error distribution often demonstrates insufficient predictive performance when analyzing proportional data. Histograms for the outcome variable must be provided. In light of the results of descriptive statistics and Figure 1, I suspect that the response variable appear to follow a bimodal distribution, accumulating many observations in both tails. To address this difficulty, I propose a beta regression model with shape parameters close to 0.5 or an asymmetric logistic regression model that uses a new parameter to account for data complexity.

RESPONSE: As described in the methods section, the primary outcome was the number of days spent at home, adjusted to a percentage. Cases were asymmetrically distributed on a 0-100 scale, with skewness towards 100% (i.e., 180 days spent at home). In light of the variable distribution, we considered the generalized linear model more likely to raise differences between factors. Our analysis could also be addressed using a zero-inflated negative binomial approach. However, we ruled out this option because of two main reasons: (1) our distribution had an upper limit, and (2) we did not have prior information regarding the variables with a strong contribution to zero in the primary outcome. As suggested, we have added the histograms of the outcome variables in the supplementary file (Figure S1).

Second, authors should provide more information to try to convince the reader that the socioeconomic status (based on annual income categories defined for pharmaceutical copayment), has sufficient explanatory power since it accumulates most of its observations in the annual income <18,000€ interval.

RESPONSE: We agree with the reviewer that the predefined cutoffs of this stratification system constrain the adjustment for socioeconomic status. Nevertheless, this stratification has been used in various studies, with consistent results that are in line with trends observed elsewhere. The limited exhaustivity of socioeconomic measurements at the population level is a common issue in many studies in the public health area (actually, few countries systematically collect the socioeconomic status or a proxy of this variable). We agree that, despite being a common issue, it must be identified as a study limitation so that the reader can appraise the accuracy of the measure. In the revised manuscript, we have addressed this issue (Discussion section, limitation paragraph) and cited other references in which the same stratification system was used.

Reviewer 2

The manuscript seems to have used sound methodology and the results and conclusions were presented in a clear way. Study limitations were also dully acknowledged. I have no suggestions or corrections to propose.

RESPONSE: We thank the reviewer for his/her appraisal of our work and hope the changes made in response to the comments of the other reviewers will suit as well.

Reviewer 3

Dear authors.

I think this is an interesting paper with a large potential. It focuses on the number of days that people in the general population (selected based on McNamara) in Catalonia, Spain, spend at home during their end-of-life, defined as 6 months before death. The paper is policy relevant and contributes nicely to a scarce literature on end-of-life trajectories.

The authors find that individuals with chronic, advanced, and progressive illnesses spend a remarkable amount of time in health and social care facilities within the last six months of life. They also find that gender and socioeconomic status influences this outcome, and therefore suggest that gender and social inequalities, often observed in healthcare provision, may also affect the end-of-life trajectory.

The paper is well written; however, it would benefit from having an English edit. The setting (i.e., how healthcare is provided in Catalonia), material (how and why registers are informed with information, and their coverage rate) is not explained well enough. The methods are quite easy to follow, however, I would prefer that the outcome of all analyses is ‘days at home’. Also, there should be a clear distinction between descriptive statistics, and analyses where the authors have predicted how factors influence the number of days spent at home while controlling for other factors.

Below, I give a point by point list of comments to the manuscript:

1. Title: I do not like the wording in the title. Maybe the title could be changed to: “Determinants of the number of days people in the general population spent at home during their end-of-life: results from a population-based cohort analysis”

RESPONSE: We agree that the proposed title is clearer and more precise. We have changed it as suggested.

2. I think the language in the abstract can be approved. For example, in the sentence: “The analysis included adult (i.e., ≥18 years) individuals died in Catalonia in 2017, who met the McNamara criteria for palliative care” I would remove ‘i.e.’, and included ‘who’: “The analysis included adult (≥18 years) individuals who died in Catalonia in 2017, who met the McNamara criteria for palliative care”.

Similarly, several other places, I think the manuscript can be improved by editing the language.

RESPONSE: We have removed the abbreviation “i.e.” and included the pronoun “who”, as suggested. We have removed the second “who” to prevent the reader from understanding that all individuals who died in Catalonia met the McNamara criteria. The revised sentence is as follows: The analysis included adult (≥18 years) individuals who died in Catalonia (North-east Spain) in 2017 and met the McNamara criteria for palliative care. We have also revised the language in the abstract. However, notice that owing to the word count limit established by the journal often challenges the exhaustivity and accuracy of the text. We have also introduced minor changes throughout the manuscript text to improve the language.

3. In the abstract, it is not intuitive to understand what the authors refer to when they indicate the results from their analyses – since the methods have not been explained. For example: “Higher socioeconomic levels were significantly associated with an increasing number of days at home in both genders: among women, ORs of the second, third, and fourth levels were 1.09 (0.97–1.22), 1.54 (1.36–1.75), and 2.52 (1.69–3.75) (p<0.001), respectively; the corresponding ORs among men were 1.27 (1.12–1.43), 1.56 (1.38–1.77), 2.82 (2.04–3.88) (p<0.001).”

What does the second, third and fourth level refer to? No levels have been introduced to the reader. Either the methods section needs to be expanded – or the results should not refer to numbers and analyses the reader can not understand how were calculated. Also, because methods have not been explained, it is impossible to understand whether the analyses in the abstract have been controlled for confounding factors or whether they are purely descriptive analyses.

As I will point out later in my comments, I think results should always be presented as ‘days at home’ – not odds ratios. This would greatly ease the read for the audience.

RESPONSE: As commented before, the word count limit established by the journal challenged the trade-off between a results section that is large enough to reflect the main findings and the methods that are necessary to understand these findings. In the revised version of the manuscript, we have expanded the methods section so that the reader can easily understand the reported results. Also, socioeconomic levels are referred to as low, mid, and high instead of first, second, and fourth. For consistency, we have included this wording in the methods section of the body text. The comment regarding the ORs is addressed below.

4. The conclusion in the abstract is strange, since it states that “the presence of gender”. Since we all have a gender, you should refer to ‘females’ or ‘males’, or that gender (not the presence of gender) influenced the results.

RESPONSE: In our original writing, the clause “the presence of” referred to “inequalities” (i.e., the presence of gender and socioeconomic inequalities). We have rephrased the conclusions section for more clarity, which now specifies also the sense of these inequalities. The current text is as follows: “Our results suggest that end-of-life care is associated with gender and socioeconomic inequalities; women and individuals with lower socioeconomic status spend less time at home within the last 180 days of life”.

5. Study design – overall – I think the material and methods are not explained well enough. See points below for details.

6. It should be stated clearly whether all adults who died in Catalonia were included?

RESPONSE: We selected individuals who met the McNamara criteria for palliative care among all deaths in Catalonia during the study period. We have rephrased the study design section for more clarity.

7. It must be elaborated what the authors have done to select the patients – what does it mean that they have used the framework by McNamara? Who have then been included? Maybe provide a patient flow?

RESPONSE: The mortality registry was screened for adult individuals who died in 2017 from any of the illnesses considered in the McNamara framework for defining palliative care populations. We believe that the changes introduced in response to comment 6 improve the clarity regarding subject selection. Furthermore, in the revised version of the manuscript, we have included a patient flow so that the reader can appraise the representativeness of our analysis (Figure S2, Supplementary file).

8. The authors should explain the healthcare system in Catalonia (public versus private, funding – tax or out of pocket) for the reader to understand the system where analyses have taken place.

RESPONSE: We agree that this feature is country-specific, and it is essential to understand the context of the analysis. In the revised version of the manuscript, we have expanded this information (Methods, Study design).

9. The authors should give a more detailed description of the data – how and why are they gathered in the registries. Do they include all healthcare services that has been provided (also privately provided services – if available)? Some of the information in the appendix can be elaborated and moved to the main manuscript?

RESPONSE: We agree that the way the healthcare system collects and stores information is relevant to understand the accuracy and exhaustivity of the data analyzed. This is particularly relevant in our health service, which provides universal health care to the entire population and links information from all datasets through a unique insurance number. In the revised manuscript, we have expanded information regarding data sources (Methods, Variables and Data Sources).

10. Why do you use a generalized linear model with a logit link and a binomial family – how did you do the model selection? I would like to know how robust these analyses are – meaning – how much are the results influenced by specifying the model in a different way?

RESPONSE: We selected the model that, to our understanding, fits better to our primary outcome (i.e., the percentage of days spent at home). In the revised version of the supplementary material, we provide the histogram corresponding to the primary variable (i.e., days spent at home in percentage) (Figure S1). This variable takes values from 0 to 100 with a marked skewness towards 100. Although other distributions might be considered (e.g., negative binomial, either zero-inflated or not) we ruled them out because the forced upper limit (to 100) might bias parameter estimate. One way to accomplish that predicted values fall between 0 to 100 is to use a generalized linear model with a logit link and the binomial family including robust option to obtain robust standard errors.

11. In the methods, what does ‘each category’ refer to?

RESPONSE: We used the term “category” to make the difference between a variable (e.g., age, socioeconomic status) and a category (e.g., 55-64 years, <€18,000), which is the actual concept associated with the OR. To make this point clearer, we have changed to “factors”, which is the term used throughout the manuscript (Methods, Analyses).

12. I think it would be a good idea to use ‘days spent at home’ as the outcome measure, also in the generalized linear models, and to select an appropriate model from among possible models, for example, the poisson, the negative binomial model, two-part models, zero-inflated models etc. See for example: Deb P, Norton EC, Manning WG. Health Econometrics Using Stata. 1 ed. Texas: Stata Press. 2017.

RESPONSE: As discussed previously, we think the generalized linear models fit better with the distribution and characteristics of our primary variable. Regarding the outcome measure, we provide the two values: the adjusted/unadjusted percentage of time (Table S3 and S4) and the OR (Figure 3). While the number of days provides the reader with a quantitative measure to appraise the extent of the differences between individuals with a given factor, we think the adjusted model is better represented by the OR, which provides an idea of the weight or contribution of each factor to explaining the primary outcome. To our understanding, this is particularly relevant for multivariate approaches like ours, in which multiple factors are affecting the primary outcome. As discussed previously, the generalized linear model is recommended when values of the dependent variable must fit between 0 and 100 (https://stats.idre.ucla.edu/stata/).

13. I really like Figure 1! Figure 2 is not possible for me to read (the picture is not clear). However, the idea of seeing where people move from and to is intriguing, so I encourage a figure showing transition.

RESPONSE: Indeed, showing transitions provide interesting information regarding end-of-life care pathways. The large sample size (more than 40,000 individuals) challenges building a meaningful and clear plot summarizing these transitions. Owing to the small number of transitions that occurred before day 171 of the investigated period (i.e., 9 days before death), we focused on this very last period. Still, we agree with the reviewer that the figure might be confusing or difficult to read. In the revised version of the manuscript, we have expanded the description of this figure (Results, Days spent at home) to facilitate interpretation. The image resolution has also been improved.

14. In the results where authors write “Overall, women and men spent a mean (SD) of 106.8(71.5) and …” it is not clear to me whether this is non-adjusted or adjusted analyses. It needs to be stated clearly!

RESPONSE: These results correspond to Table 2, which presents the description of the untransformed primary variable (i.e., number of days spent at home). As explained in the section “Methods, Analyses”, the number of days spent in each place was first described using the mean (SD) and the median (IQR) and then transformed to percentage to build the models (adjusted and unadjusted) aimed to investigate the influence of each factor. We agree that this approach was not clear enough in the previous version of the manuscript, and we have revised the methods section to make this clearer. However, we think including the term “unadjusted” when mentioning Table 2 in the text may be misleading because the reader may expect an adjusted analysis of the number of days spent in each place (in contrast, the number of days was used only for descriptive purposes). Instead, we have reworded the first paragraph of the section “Methods, Analyses”, the first sentence of the section “Methods, Variables and Outcomes”, and the heading of Table 2, which now explicitly state the descriptive nature of the results presented. We think that these changes, along with the description of adjusted vs. unadjusted results included as response to comment 16, improve clarity of the text.

15. The analyses of factors influencing time spent at home should be analyzed with the output ‘days spent at home’. For me, it is not intuitive to understand how the OR and percentage of days spent at home should be interpreted. However, if you give results in the no. of days spent at home (with reference either to a reference category, or, as the total no. of days) it would be much easier to read.

RESPONSE: As discussed in response to comment 12, we presented both the adjusted/unadjusted percentage of days spent at home (Table S4 and S5) and other places (Tables S1 and S2), and the OR associated with each factor (Figure 3). To our understanding, the OR provides the reader with an idea of the relative weight (and statistical significance) of the contribution of each factor to the variability in the primary outcome.

16. The authors should elaborate on what changes when they run crude and fully controlled analyses.

RESPONSE: In the revised version of the manuscript, we have provided this information in-text and cited Table S4, which shows the results of the unadjusted analysis.

17. Is there a difference pattern of end-of-life trajectories between those dying from different causes – can the analyses be stratified by cause of death (as it has been by gender)?

RESPONSE: We agree that an in-depth analysis of the trajectories according to causes of death would have high value for making decisions regarding the healthcare pathways of these patients. This analysis is challenged by the way the healthcare system deals with the end-of-life of each of these diagnoses and how grouping them to make sense of the analysis results. We think this analysis should be addressed in future work; therefore, in the revised version of the manuscript, we have included a statement in this regard (Conclusions).

---

## [Decision Letter · Decision Letter 1]

7 Jun 2021

Determinants of the Number of Days People in the General Population Spent at Home During End-of-Life: Results from a Population-Based Cohort Analysis

PONE-D-20-40096R1

Dear Dr. Amblàs,

We’re pleased to inform you that your manuscript has been judged scientifically suitable for publication and will be formally accepted for publication once it meets all outstanding technical requirements.

Kind regards,

Rosa Maria Urbanos Garrido, PhD

Academic Editor

PLOS ONE

Reviewers' comments:

Reviewer's Responses to Questions

**Comments to the Author**

1. If the authors have adequately addressed your comments raised in a previous round of review and you feel that this manuscript is now acceptable for publication, you may indicate that here to bypass the “Comments to the Author” section, enter your conflict of interest statement in the “Confidential to Editor” section, and submit your "Accept" recommendation.

Reviewer #1: All comments have been addressed

Reviewer #3: All comments have been addressed

2. Is the manuscript technically sound, and do the data support the conclusions?

Reviewer #1: Yes

Reviewer #3: Yes

3. Has the statistical analysis been performed appropriately and rigorously? 

Reviewer #1: Yes

Reviewer #3: Yes

4. Have the authors made all data underlying the findings in their manuscript fully available?

Reviewer #1: Yes

Reviewer #3: No

5. Is the manuscript presented in an intelligible fashion and written in standard English?

Reviewer #1: Yes

Reviewer #3: Yes

6. Review Comments to the Author

Reviewer #1: The authors have responded correctly to most of my comments, and they have revised article accordingly.

Reviewer #3: The authors have addressed my comments to their paper, from my first review, in a sufficient manner. When reading the manuscript now – I think it has improved and is a lot easier to read and understand. I therefore have no further comments.

7. PLOS authors have the option to publish the peer review history of their article (what does this mean?). If published, this will include your full peer review and any attached files.

Reviewer #1: No

Reviewer #3: **Yes: **Gudrun Waaler Bjørnelv

---

## [Editor Report · Acceptance letter]

24 Jun 2021

PONE-D-20-40096R1 

Determinants of the Number of Days People in the General Population Spent at Home During End-of-Life: Results from a Population-Based Cohort Analysis 

Dear Dr. Amblàs:

I'm pleased to inform you that your manuscript has been deemed suitable for publication in PLOS ONE. Congratulations! Your manuscript is now with our production department. 

Kind regards, 

on behalf of

Dr. Rosa Maria Urbanos Garrido 

Academic Editor

PLOS ONE